# Suspected Primary Intracranial Melanoma with Widespread Distant Metastases in a Cat

**DOI:** 10.3390/ani13243751

**Published:** 2023-12-05

**Authors:** Jonathan Deacon, Samuel Beck, Francesca Pitorri, Catherine Stalin

**Affiliations:** 1Moorview Referrals, Northumberland Business Park West, Newcastle NE23 7RH, UK; jonathand@moorviewreferrals.co.uk; 2Independent Anatomic Pathology Ltd., Calyx House, South Road, Taunton TA1 3DU, UK; 3VPG Leeds, Unit 8, Temple Point, Bullerthorpe Ln, Colton, Leeds LS15 9JL, UK; 4School of Biodiversity, One Health and Veterinary Medicine, University of Glasgow, Garscube Campus, Bearsden Road, Glasgow G61 1QH, UK

**Keywords:** feline, melanoma, intracranial, histopathology

## Abstract

**Simple Summary:**

This report describes the diagnosis of an intracranial melanoma in an 8-year-old female neutered Domestic Shorthair cat. Melanomas are one of the few lesions that present as hyperintense on T1-weighted sequences and have only previously been reported intracranially as an extension of an ocular mass. This case supports the diagnosis of primary intracranial melanoma in a cat and establishes the importance of melanoma as a differential for intracranial neoplasia.

**Abstract:**

An 8-year-old female Domestic Shorthair presented with signs of intracranial disease. Magnetic resonance imaging (MRI) of the head showed an extra-axial space-occupying mass within the cranial vault with a similar intensity lesion within the overlying temporalis muscle. Postmortem examination found masses within the head, lung, liver, spleen, and kidney consistent with malignant melanoma. Intracranial melanoma is rarely reported in cats and is typically only seen as a metastatic lesion associated with an ocular mass. Melanomas can be readily recognised on MRI as they are one of the few lesions which are hyperintense on T1-weighted images.

## 1. Introduction

Melanomas arise from the uncontrolled proliferation of melanocytes found within specific tissues reflecting the different subtypes: cutaneous, mucosal, ocular, and leptomeningeal [1]. Melanoma at any site in the cat is uncommon [1], and it has been recognised that different from other species, ocular melanomas are more common than oral or cutaneous melanomas [2]. Central nervous system melanomas have previously been reported in cats but only as directly invasive ocular lesions or distant spinal metastases [3]. In humans, the CNS can be affected by various melanocytic lesions ranging from benign melanocytomas to malignant metastatic melanomas [4]. Rarely, the intracranial presence of melanin is associated with primary leptomeningeal melanoma either as a type diffusely invading the pia matter and spreading into the subarachnoid space or as a type forming nodular tumours [4,5].

Magnetic resonance imaging is the modality of choice for diagnosing a lesion localised intracranially. Most pathologies will present hypo-intense on T1-weighted sequences. However, there are a small number of substances and pathologies that present as T1 hyperintense, and knowledge of this, along with localization, can assist in narrowing a diagnosis [5]. Paramagnetic substances such as gadolinium contrast and methaemoglobin will appear hyperintense on T1-weighted images along with certain minerals (calcium, copper, iron, and manganese), fat-containing, and protein-rich substances. Melanin also has paramagnetic properties and will exhibit high intensity on T1-weighted images and low intensity on T2-weighted images.

In this case, a large space-occupying T1-hyperintense and T2-hypointense lesion was identified intracranially and in the adjacent temporal muscle of a cat first presenting with neurological disease. Cytology confirmed the lesion as melanoma, and subsequent postmortem examination further defined the intracranial mass and identified randomly distributed smaller lesions within the spleen, liver, lung, and kidney consistent with distant metastasis.

## 2. Case Presentation

An 8-year-old spayed female Domestic Shorthair was referred for neurological assessment as she had been reluctant to move for three days following the administration of a single 20 mg/Kg dose of gabapentin. She had a three-week history of reluctance to jump, changed sleeping habits, pain on lumbar palpation, and a two-week history of ataxia and weakness in the pelvic limbs. Radiographs of the lumbar spine, pelvis, and stifles from the referring veterinarian were reported as normal. The patient’s general physical examination, including oral examination and palpation of the lymph nodes and skin surface, was unremarkable other than being very obese. Neurological assessment, including ophthalmologic examination, indicated mild obtundation, non-ambulatory tetraparesis, and absence of reduced conscious proprioception (left limbs worse than right). Spinal reflexes and cranial nerve examination were normal. Any movements attempted appeared exaggerated and uncoordinated, and she was not able to right herself properly from either lateral recumbency. The complete blood count and serum biochemistry were unremarkable.

## 3. Diagnostic Imaging

To determine the cause of the neurological signs, an MRI of the brain was performed using a 1.5T magnet (Toshiba Vantage Elan, Canon medical systems Ltd., Crawley, UK). Sequences included T2-weighted (T2W) in the sagittal (TR 5000, TE 96, 2.5 mm) and transverse (TR 8487, TE 96, 2.5 mm) planes, transverse T2W fluid attenuated inverse recovery (FLAIR; TR 11000, TE 108, 2.5 mm) and transverse T1-weighted (T1W; TR 490, TE 10, 2.5 mm) before and after the administration of contrast (Gadovist, Bayer, Ontario Canada dose of 0.1 mL/Kg = 278.2 mg gadobutrol). These showed an extra-axial space occupying a mass lesion on the left side of the cranial vault, compressing the underlying parietal and occipital lobes and causing a midline shift (Figure 1). There was also marked caudal subtentorial and foraminal herniation. The mass was hypointense on T2W and FLAIR and hyperintense on T1W images. An infiltration of similar intensity was also seen within the muscle on the extracranial side of the parietal bone.

## 4. Cytology, Postmortem and Histopathology

A fine needle aspirate of the extracranial lesion showed cells in an eosinophilic background containing a large number of small blue/black irregularly shaped fine granules similar in appearance to melanin. Nucleated cells comprised a pleomorphic population of round to polyhedral cells exhibiting marked anisokaryosis and anisocytosis, additionally containing blue–black granules as seen in the extracellular space (Figure 2). The findings were consistent with melanoma.

Due to the poor prognosis, the patient was euthanised prior to any further antemortem tests, and a postmortem was performed. Within the cranium, an extra-axial irregularly shaped, soft, friable, and nodular dark brown–black mass measuring approximately 50 mm in diameter was present contiguous with the frontal bone and meninges. The mass extended through the leptomeninges lining the dorsal cerebral hemispheres and cerebellum, obscuring up to 80% of their dorsal area. The leptomeningeal lesions were coalescing, soft, indistinct, flat, and plaque-like, obscuring the underlying neuroparenchyma (Figure 3). Nodular to plaque-like dark brown–black mass lesions were present, effacing approximately 60% of the frontal and parietal bones, extending through them, and forming smaller plaque-like foci infiltrating approximately 10% of the occipital bone. Nodular, soft, variably sized (1–10 mm) similarly coloured lesions were grossly identified and randomly distributed throughout the lungs, kidneys, liver, and spleen. The occipital/parietal bone, extra-axial mass, brain, lung, kidney, liver, and spleen were sampled and submitted for histopathology.

Histopathology of the extra-axial mass identified a moderately well-demarcated, non-encapsulated, densely cellular neoplasm infiltrating and replacing a dense fibrous connective tissue stroma, forming sheets, cords, and nests. Individual neoplastic cells were pleomorphic, predominantly plump spindloid, to round to less frequently polygonal with abundant eosinophilic cytoplasm containing large numbers of brown–black pigment granules (melanin) and a single nucleus. Nuclei were round to oval, with a single prominent nucleolus and stippled chromatin. There was moderate to marked anisocytosis and anisokaryosis, with less than one mitosis identified in 10 high power fields (2.37 mm^2^). The neoplasm was identified as infiltrating the overlying dura mater, bone, and skeletal muscle. There was a neoplasm with similar histologic features to those previously described, bluntly infiltrating and expanding leptomeningeal layers overlying the cerebral cortices, cerebellum, ventral pons, and medulla. The neoplastic population did not infiltrate beyond the leptomeninges into the subjacent brain parenchyma that otherwise appeared well organised (Figure 4a,b).

Within the liver, multifocally and randomly, hepatocytes were separated by clusters and nests of the neoplastic population as previously described (Figure 4c). The pulmonary interstitium was multifocally and randomly expanded and infiltrated by the neoplastic population (Figure 4d). In the spleen, there was more extensive replacement by coalescing sheets of the neoplastic population. In the kidney, there was an infiltrative, well-demarcated, non-encapsulated neoplasm with similar histologic features to those previously described.

A diagnosis of primary meningeal melanoma with widespread metastases was made.

## 5. Discussion

Melanocytic neoplasia is considered uncommon to rare in cats [6,7] in comparison to dogs and humans. Although the eye is considered to be the most commonly involved site, non-ocular melanomas have been reported [2,6,7,8] with a propensity for cutaneous tumours to present in the head. Although in humans, patients with primary lesions of the head and neck or oral mucosa have a higher incidence of central nervous system (CNS) metastases, there are few reports of intracranial melanoma in dogs [9,10] and cats [2,11].

This case is unusual as no cutaneous, ocular, or mucosal primary tumour was identified both on physical examination and magnetic resonance imaging of the eyes and optic tracts, and clinical diagnosis was made based on the neurological deficits, characteristic magnetic resonance imaging findings of a hyper-intense lesion on T1-weighted images and cytology.

Based on our findings, the authors interpret the primary site of oncogenesis to be within the leptomeninges; this is not unreasonable given that pigmented cells derived from the neural crest have been identified in the leptomeninges of the cat [12] and that primary CNS melanoma in humans is derived from the melanocytes of the leptomeninges [13]. The tumour, in this case, was predominantly within the leptomeninges, with no infiltration into the adjacent brain parenchyma supporting this interpretation.

Previous descriptions of CNS melanoma in the cat have described multiple infiltrative dark masses [10], more indicative of metastatic spread as compared to the lesion observed in this case. Primary CNS melanoma is uncommon in humans [13], and there is only a single report in a dog where a primary CNS melanoma has been suggested [9]. A recent case series of non-ocular melanoma in 324 cats did not describe a single intracranial case [7]. It has been recognised that tumour cells spread to the leptomeninges by arterial or venous circulation, intimate contact with the tumour, or centripetal progression along perineural spaces [14]. Metastasis to this intracranial site was therefore considered; however, it is thought less likely given the relatively large mass noted consistent with a primary site of oncogenesis. Metastasis was also not identified within the brain that would otherwise support spread to the cranium. The randomly distributed pattern of embolic seeding of the remaining organs is consistent with metastasis to these sites rather than primary neoplastic transformation.

In humans, it is difficult to recognize the difference between a primary malignant melanoma in a non-cutaneous site and a metastatic lesion [13,15]. It has been suggested that CNS malignant melanoma is more likely to be primary if there is a single intracranial lesion, a lesion with intramedullary or leptomeningeal involvement, a lesion in the pituitary or pineal gland, a lesion with hydrocephalus or a lesion without malignant melanoma outside the CNS. The latter is suggested as it is rare for malignant tumours to metastasize outside the head [16]. However, other sources recognizing the diagnostic difficulty of diagnosing a primary meningeal melanoma suggest that the exclusion of a primary cutaneous or mucosal/retinal melanoma is sufficient for primary CNS melanoma diagnosis [4].

In this case, with multiple melanomas identified outside the CNS, it is still difficult to describe this single large leptomeningeal mass as a primary CNS melanoma. However, the other melanomas identified are equally unlikely sites for the primary lesion, and no lesions were identified in the more common ocular or cutaneous sites. Melanoma has been reported in the temporalis muscle of a human patient, but as a metastatic melanoma of unknown primary (MUOP) and other MUOPs have been described [17]. Primary pulmonary melanoma is rarely diagnosed and is only considered if there is a solitary lung tumour with no previous ocular or skin tumours [15]. In this case, the multifocal and random distribution of neoplastic populations within the lungs, liver, and spleen are more indicative of metastases. While secondary malignant melanoma of the kidney is not uncommon, primary melanomas are extremely rare, so it is unlikely that the infiltrative mass in the kidney was the primary melanoma.

There are other unusual features of the histopathology in this case. At the presumed site of primary neoplastic transformation, there is an absence of local neuroparenchymal infiltration; this is somewhat unexpected given the extensive local infiltration of the overlying frontal/parietal bone, although potentially local seeding of the underlying brain would have occurred with time. Depending on the site, a recently described grading scheme [7] identifies a mitotic count of 4 or greater in 10 high-power fields (2.37 mm^2^) and/or intratumoral necrosis as consistent with a high-grade tumour. This designation has been associated with a reduced median survival time. Interestingly, this neoplasm had neither an elevated mitotic count nor intratumoral necrosis so it would have been low-grade; this highlights the challenges in prognostication of melanocytic tumours in general, which has been extensively discussed in canine patients [18]. The site of feline non-ocular melanoma was determined as significant for prognosis similar to the situation in canids [7,18]. Whilst similar cases have not been reported to our knowledge, an intracranial location of melanocytic neoplasia will very likely be associated with a poor outcome. The widespread metastasis is also unusual and would not necessarily have been expected in the context of the location, dense pigmentation, and low mitotic count.

## 6. Conclusions

It is likely that this case represents a primary intracranial melanoma, although the diagnostic difficulty of differentiating between primary and metastatic origins must be recognised. It is, therefore, important to recognize that melanoma is a potential differential for an intracranial mass in cats and that it can be readily recognised on magnetic resonance images due to the unusual hyperintensity on T1 images, which is not typically seen with other neoplasms.

## Figures and Tables

**Figure 1 animals-13-03751-f001:**
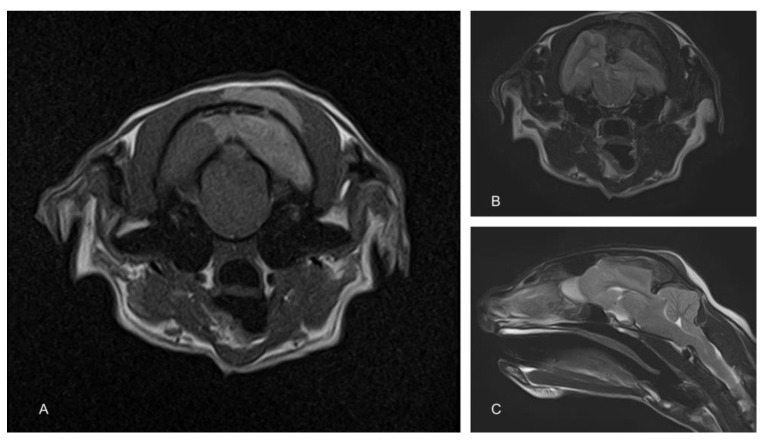
Magnetic resonance images of the cranium. The T1-weighted image shows a hyperintense extra-axial mass compressing the left cerebral hemisphere. A similar hyperintense mass is also apparent in the overlying muscle (**A**). T2-weighted images show a similarly positioned hypointense mass (**B**). Compression of the cerebrum results in significant herniation of the cerebellum through the caudal fossa (**C**).

**Figure 2 animals-13-03751-f002:**
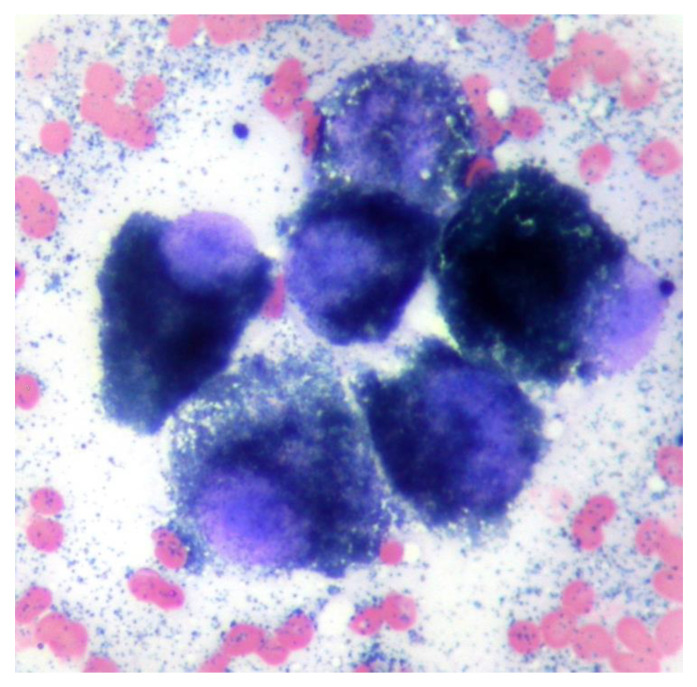
Cytology from a fine needle aspiration of the mass shows a group of round to polyhedral cells with single large round to oval nuclei with prominent single nucleoli and intracytoplasmic small blue/black melanin granules. Modified Wright’s ×100 objective.

**Figure 3 animals-13-03751-f003:**
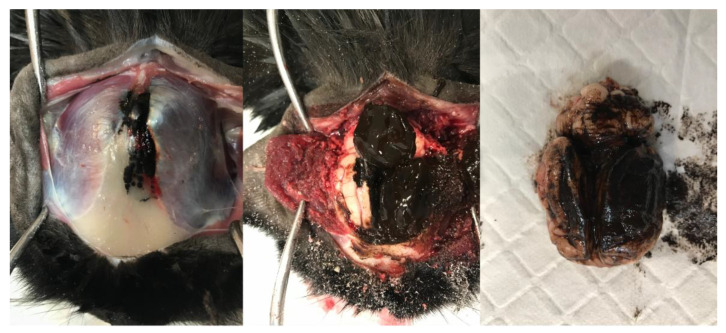
Sequential gross images illustrating dissection of the head and brain removal. A multifocal, soft, indistinct, flat to plaque-like brown–black lesion is present within the frontal and occipital bones and temporalis muscle. The lesions are contiguous with a large, soft, similarly coloured mass within the cranium and leptomeninges, obscuring the dorsal cerebral hemispheres, cerebellar vermis, and hemispheres.

**Figure 4 animals-13-03751-f004:**
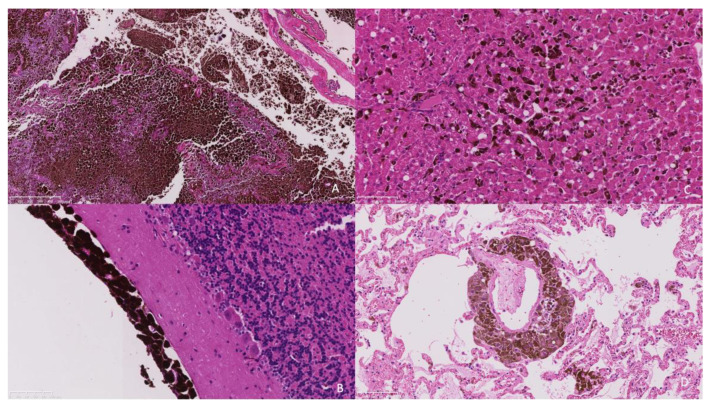
(**A**). The frontal bone is infiltrated and replaced by an infiltrative pigmented neoplastic population (×5 objective). (**B**). The neoplastic population is present within the leptomeninges overlying the cerebellum without extension into the underlying neuroparenchyma (×20 objective). (**C**). The hepatic sinusoids contain randomly distributed neoplastic cells suggestive of embolic spread. (×20 objective). (**D**). Similarly, the pulmonary interstitium contains randomly distributed tumour emboli. (×20 objective).

## Data Availability

The data presented in this study are available in the article.

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
