# Peer review of "Suspected Primary Intracranial Melanoma with Widespread Distant Metastases in a Cat"

_animals, 2023, doi:10.3390/ani13243751_

Round 1
Reviewer 1 Report
Comments and Suggestions for Authors
General concept comments
- The case described is interesting due to the rarity of localization of intracranial melanocytic tumors, particularly in cats, which do not derive from ocular or oral melanomas.
- The case is well documented; intracranial lesions can be described in more detail (shape, size) also because in the photo the black color covers all the macroscopic details (a macro image of the brain in cut section could be added)
- The references are not very up to date. The bibliography is scarce in this specific case but at least in the general parts more recent references can be included (e.g. El Ouazzani H, Oudghiri MY, Abbas S, Regragui A, Elouahabi A, Zouaidia F, Cherradi N. Diagnostic challenge: primary leptomeningeal melanoma with melanomatosis, illustrative case report. J Surg Case Rep. 2023 Jun 12;2023(6):rjad323. doi: 10.1093/jscr/rjad323. PMID: 37313430; PMCID: PMC10260324. Fert S, River P, Bondonny L, Cauzinille L. Metastatic extradural melanoma of the lumbar spine in a cat. Vet Med Sci. 2023 Sep 1. doi: 10.1002/vms3.1248. Epub ahead of print. PMID: 37656442. van der Weyden L, Brenn T, Patton EE, Wood GA, Adams DJ. Spontaneously occurring melanoma in animals and their relevance to human melanoma. J Pathol. 2020 Sep;252(1):4-21. doi: 10.1002/path.5505. Epub 2020 Jul 31. PMID: 32652526; PMCID: PMC7497193.)
- The title of the article in my opinion is too generic and suitable for a review on melanocytic tumors in cats. It is suggested to change the title in more detail with respect to the topic addressed.
- It is not clear for me why the authors did not confidently define the tumor as primary meningeal melanoma since through instrumental examinations and after complete necropsy it was not possible to prove the primary onset of the melanoma in other more common locations; I would suggest that the authors emphasize this concept and the supporting evidence more convincingly. I suggest the authors integrate the discussion with the possible differential diagnoses of intracranial neoplasia of meningeal origin (e.g. meningioma, nerve sheath tumor). To define the nature of the neoplasm more precisely and definitively exclude other differential diagnoses, I would perform immunohistochemical investigations, after discoloration, of a fragment of the neoplasm.
Specific comments
Line 7: to which author does note 4 refer?
Line 36: add suggested reference (Ouazzani et al, 2023).
Lines 86-89: instead of the word “lesion” use a more descriptive term (mass, plaque, nodule, macula, etc.). Details the description of the meningeal mass (size, shape) and the calvarium involved (thickness, etc.). Describe better the lesions in the other organs (distribution, size, type of lesion). Were no lymph nodes in the anatomical regions affected by the tumor metastatic? Here I would like to underline that no ocular or mucosal lesions of the oral cavity were found during the necropsy examination.
Line 90 figure 3: add macroscopic picture of cross section of the brain.
Line 96: describe, if present, the prevalent cellular morphology of the neoplastic cells (epithelioid, round, spindle).
Line 102: Were all the meninges involved? Detail.
Line 119: I would be more precise in the localization of the neoplasm (meningeal melanoma).
Lines 122-124: add more updated bibliography.
Line 200: enter the reference correctly (authors in full, title of the article, etc.).
Author Response
Thank you for taking the time to review our manuscript. The points you have made have been very useful in allowing us to clarify and expand our case report and I believe it is now very much improved. Please find a response to all your observations below:
- The case described is interesting due to the rarity of localization of intracranial melanocytic tumors, particularly in cats, which do not derive from ocular or oral melanomas.
- The case is well documented; intracranial lesions can be described in more detail (shape, size) also because in the photo the black color covers all the macroscopic details (a macro image of the brain in cut section could be added)
Following review of the available images taken during necropsy at the hospital one author (SB) has provided additional description. We do not have images of gross cut sections of the brain. The lab did not routinely photograph brain at the time. We do have photomicrographs showing the infiltration of the meninges however, which matches the gross and histologic descriptions. A composite gross image has been provided in place of the original figure 3.
- The references are not very up to date. The bibliography is scarce in this specific case but at least in the general parts more recent references can be included (e.g. El Ouazzani H, Oudghiri MY, Abbas S, Regragui A, Elouahabi A, Zouaidia F, Cherradi N. Diagnostic challenge: primary leptomeningeal melanoma with melanomatosis, illustrative case report. J Surg Case Rep. 2023 Jun 12;2023(6):rjad323. doi: 10.1093/jscr/rjad323. PMID: 37313430; PMCID: PMC10260324. Fert S, River P, Bondonny L, Cauzinille L. Metastatic extradural melanoma of the lumbar spine in a cat. Vet Med Sci. 2023 Sep 1. doi: 10.1002/vms3.1248. Epub ahead of print. PMID: 37656442. van der Weyden L, Brenn T, Patton EE, Wood GA, Adams DJ. Spontaneously occurring melanoma in animals and their relevance to human melanoma. J Pathol. 2020 Sep;252(1):4-21. doi: 10.1002/path.5505. Epub 2020 Jul 31. PMID: 32652526; PMCID: PMC7497193
Thank you for providing these additional references. The introduction has been expanded to give a broader description of melanoma and more references have been included.
- The title of the article in my opinion is too generic and suitable for a review on melanocytic tumors in cats. It is suggested to change the title in more detail with respect to the topic addressed.
The title has been changed as suggested.
- It is not clear for me why the authors did not confidently define the tumor as primary meningeal melanoma since through instrumental examinations and after complete necropsy it was not possible to prove the primary onset of the melanoma in other more common locations; I would suggest that the authors emphasize this concept and the supporting evidence more convincingly. I suggest the authors integrate the discussion with the possible differential diagnoses of intracranial neoplasia of meningeal origin (e.g. meningioma, nerve sheath tumor). To define the nature of the neoplasm more precisely and definitively exclude other differential diagnoses, I would perform immunohistochemical investigations, after discoloration, of a fragment of the neoplasm
The text has been changed to give a more confident diagnosis of intracranial melanoma given that on examination no ocular, cutaneous of mucosal lesion was identified and the eyes and optic nerves and tracts appeared normal on magnetic resonance imaging.The pigmentation, nuclear morphology and cellular pleomorphism are not consistent with the other described intracranial tumours. Similarly widespread metastasis would not be expected with these entities therefore immunohistochemistry was not deemed necessary for confident diagnosis. The extra cost was not considered necessary or supported by the owner.
Specific comments
Line 7: to which author does note 4 refer?
This has been corrected.
Line 36: add suggested reference (Ouazzani et al, 2023).
This has been added with a slightly altered description of primary intracranial melanoma in humans.
Lines 86-89: instead of the word “lesion” use a more descriptive term (mass, plaque, nodule, macula, etc.). Details the description of the meningeal mass (size, shape) and the calvarium involved (thickness, etc.). Describe better the lesions in the other organs (distribution, size, type of lesion). Were no lymph nodes in the anatomical regions affected by the tumor metastatic? Here I would like to underline that no ocular or mucosal lesions of the oral cavity were found during the necropsy examination
Further descriptions have been provided. Lymph nodes were not sampled by the prosecting vet as they were not enlarged or abnormal on magnetic resonance imaging. Similarly, the eyes were not necropsied as they were perceived normal on ophthalmological examination (part of neurological examination) and there were no identifiable lesions in the eye, optic nerves or tracts on MRI making iris or uveal melanoma unlikely.
The oral cavity and dermis were examined ante- and post-mortem. No mass lesion was palpated on the skin but the cat was not clipped to fully examine the cutaneous surface.
Line 90 figure 3: add macroscopic picture of cross section of the brain.
Routine diagnostic laboratory practice did not include photographing cut section of brain at the time of submission. Additional gross image supplied.
Line 96: describe, if present, the prevalent cellular morphology of the neoplastic cells (epithelioid, round, spindle).
Done
Line 102: Were all the meninges involved? Detail.
Done
Line 119: I would be more precise in the localization of the neoplasm (meningeal melanoma).
More precise diagnosis made.
Lines 122-124: add more updated bibliography.
Included suggested references with a broader description of melanoma
Line 200: enter the reference correctly (authors in full, title of the article, etc.).
References have been corrected.
Reviewer 2 Report
Comments and Suggestions for Authors
The authors describe an interesting case of possible primary intracranial melanoma. The case report is interesting and even if small add to exiting literature in this rare type of cancer in cats, however need major revision.
The title is too generic, no clear and no of the case . Please consider to change to: Suspected primary intracranial melanoma with wide spread distant metastases in a cat.
No authors have the 4th affiliation
The introduction need to add more information and literature on feline melanoma, oral, ocular etc, rather then just MRI image background that it is not the main point of the case report!
case descriptions; need to add any regional lymphadenopathy? any mass felt on the top of the head?
in the diagnostic imaging or post mortem: Any regional lymphadenopathy? This is important as melanoma metastases via lymphatic and it is the first site of metastases. especially if was from the muscle/non intracranial in origin, it would be expected to go in the regional lymph nodes first.
Why thoracic radiographies were not performed? This i snot standard of practice, at least justify..
In the post mortem macroscopic examination need some work. The "lesions in the abdomen and thorax " need to be better explained. Are you talking about pigmented nodules? where exactly? liver, kidneys, lungs?
Was the eyes examined and ruled out as a primary? did any section of the eyes examined microscopically? did the oral cavity and the skin examined to rule out any small primary? please add all these info.
In the discussion it should be mentioned that the mitotic index was quite low for such an aggressive cancer and compare to other reports of aggressive melanoma in cats.
It is also important discussing that if the origin of the melanoma is the meninge as was suggested it is extremely unusual to invade the bone and the temporal muscles but not the brain. have you ever seen meningioma doing that? need to be discussed.
Minor comments
Line 40 "small lesions" is this just pigment flat lesions or actual masses/nodules?
Line 74. the title is not appropriate as describe the macroscopic post mortem , cytology and histopathology, add all
Line 86 "lesion" please specify what lesion is? a mass? hard soft, this is standard for macroscopic description at post mortem. Please ask the pathologist to describe better!
Author Response
Thank you for taking the time to review our manuscript. The comments have been very useful to improve the description and discussion on this case. Please see the responses to all comments below:
The authors describe an interesting case of possible primary intracranial melanoma. The case report is interesting and even if small add to exiting literature in this rare type of cancer in cats, however need major revision.
- The title is too generic, no clear and no of the case . Please consider to change to: Suspected primary intracranial melanoma with wide spread distant metastases in a cat.
The title has been changed as suggested.
- No authors have the 4th affiliation
This has been corrected.
The introduction need to add more information and literature on feline melanoma, oral, ocular etc, rather then just MRI image background that it is not the main point of the case report!
Introduction has been expanded along with bibliography to reflect the current literature on feline melanoma and primary intracranial melanoma.
case descriptions; need to add any regional lymphadenopathy? any mass felt on the top of the head?
Specific examination included in the physical and neurological examination has been included. This includes ocular examination, lymph node palpation and palpation of the skin surfaces.
In the diagnostic imaging or post mortem: Any regional lymphadenopathy? This is important as melanoma metastases via lymphatic and it is the first site of metastases. especially if was from the muscle/non intracranial in origin, it would be expected to go in the regional lymph nodes first.
No obvious lymphademopathy on palpation or imaging
Why thoracic radiographies were not performed? This is not standard of practice, at least justify.
The cat presented with an intracranial localisation and no physical abnormalities. It would therefore be standard practice to image the brain for a diagnosis prior to ancillary tests for metastases. The prognosis however, was so poor in this case that the additional cost to the owner was not justified to perform thoracic radiographs ante-mortem. For diagnosis, this was accounted for by examining the thoracic and abdominal cavities post-mortem.
In the post mortem macroscopic examination need some work. The "lesions in the abdomen and thorax " need to be better explained. Are you talking about pigmented nodules? where exactly? liver, kidneys, lungs?
Done
Were the eyes examined and ruled out as a primary? did any section of the eyes examined microscopically? did the oral cavity and the skin examined to rule out any small primary? please add all these info.
Physical examination included ocular and oral examination as well as palpation of lymph nodes and skin surface. Magnetic resonance imaging of the head included the eyes and optic nerve/tract as well as the local lymph nodes. This has been included in the text.
In the discussion it should be mentioned that the mitotic index was quite low for such an aggressive cancer and compare to other reports of aggressive melanoma in cats.
It is also important discussing that if the origin of the melanoma is the meninge as was suggested it is extremely unusual to invade the bone and the muscles but not the brain. have you ever seen meningioma doing that? need to be discussed.
Unusual features of this non-ocular melanoma are discussed further in text.
Minor comments
Line 40 "small lesions" is this just pigment flat lesions or actual masses/nodules?
Done
Line 74. the title is not appropriate as describe the macroscopic post mortem , cytology and histopathology, add all
This has been corrected.
Line 86 "lesion" please specify what lesion is? a mass? hard soft, this is standard for macroscopic description at post mortem. Please ask the pathologist to describe better!
Done as far as possible from records of necropsy.
Round 2
Reviewer 2 Report
Comments and Suggestions for Authors
no further comments